# A Multigene Signature Associated with Progression-Free Survival after Treatment for IDH Mutant and 1p/19q Codeleted Oligodendrogliomas

**DOI:** 10.3390/cancers15123067

**Published:** 2023-06-06

**Authors:** Julia Gilhodes, Adèle Meola, Bastien Cabarrou, Guillaume Peyraga, Caroline Dehais, Dominique Figarella-Branger, François Ducray, Claude-Alain Maurage, Delphine Loussouarn, Emmanuelle Uro-Coste, Elizabeth Cohen-Jonathan Moyal

**Affiliations:** 1Biostatistics & Health Data Science Unit, Institut Claudius Regaud, Oncopole Claudius Regaud-Institut Universitaire du Cancer Toulouse, 31100 Toulouse, France; gilhodes.julia@iuct-oncopole.fr (J.G.); cabarrou.bastien@iuct-oncopole.fr (B.C.); 2Department of Radiation Oncology, Institut Claudius Regaud, Oncopole Claudius Regaud-Institut Universitaire du Cancer Toulouse, 31100 Toulouse, France; meola.adele@iuct-oncopole.fr (A.M.); peyraga.guillaume@hotmail.fr (G.P.); 3Neuro-Oncology Department, Assistance Publique-Hôpitaux de Paris, Hôpitaux Universitaires La Pitié Salpêtrière-Charles Foix, Sorbonne University, 75006 Paris, France; caroline.dehais@aphp.fr; 4Department of Pathology, Centre Hospitalo-Universitaire Timone, AP-HM, GlioME Team, Institute of Neurophysiopathology, Aix-Marseille University, 13385 Marseille, France; dominiquefrance.figarella@ap-hm.fr; 5Neuro-Oncology Department, Hospices Civils de Lyon, Université Lyon 1, CRCL, UMR Inserm 1052_CNRS 5286, 69003 Lyon, France; francois.ducray@chu-lyon.fr; 6Department of Pathology, Lille University Hospital, 59000 Lille, France; ca-maurage@chru-lille.fr; 7Pathology Department, Hotel-Dieu, CHU Nantes, 44093 Nantes, France; delphine.loussouarn@chu-nantes.fr; 8Department of Pathology, CHU Toulouse, Institut Universitaire du Cancer Toulouse, 31100 Toulouse, France; uro-coste.e@chu-toulouse.fr; 9Centre de Recherches Contre le Cancer de Toulouse, INSERM U1037, 31100 Toulouse, France

**Keywords:** glioma, 1p/19q codeletion, gene signature, treatment response

## Abstract

**Simple Summary:**

Since the publication in 2016 of the WHO’s classification of primary brain tumors according to their histopathology but also their molecular status (IDH, 1p/19q codeletion), oligodendrogliomas defined by the presence of the 1p/19q codeletion have been clearly identified as having a better prognosis. However, the response to treatment of 1p/19q codeleted gliomas remains heterogeneous. Very few studies have investigated the genetic profiles of these tumors, particularly with regard to their response to treatment (radiotherapy and chemotherapy). Our analyses revealed a gene signature composed of eight genes involved in metabolism, immunity, and extracellular matrix organization pathways that were associated with a poor response to treatment for 1p/19q codeleted tumors. This signature could be used in the future to identify patients who need more intensive treatment, potentially with inhibitors of these pathways.

**Abstract:**

Background. IDH mutant and 1p/19q codeleted oligodendrogliomas are the gliomas associated with the best prognosis. However, despite their sensitivity to treatment, patient survival remains heterogeneous. We aimed to identify gene expressions associated with response to treatment from a national cohort of patients with oligodendrogliomas, all treated with radiotherapy +/− chemotherapy. Methods. We extracted total RNA from frozen tumor samples and investigated enriched pathways using KEGG and Reactome databases. We applied a stability selection approach based on subsampling combined with the lasso-pcvl algorithm to identify genes associated with progression-free survival and calculate a risk score. Results. We included 68 patients with oligodendrogliomas treated with radiotherapy +/− chemotherapy. After filtering, 1697 genes were obtained, including 134 associated with progression-free survival: 35 with a better prognosis and 99 with a poorer one. Eight genes (ST3GAL6, QPCT, NQO1, EPHX1, CST3, S100A8, CHI3L1, and OSBPL3) whose risk score remained statistically significant after adjustment for prognostic factors in multivariate analysis were selected in more than 60% of cases were associated with shorter progression-free survival. Conclusions. We found an eight-gene signature associated with a higher risk of rapid relapse after treatment in patients with oligodendrogliomas. This finding could help clinicians identify patients who need more intensive treatment.

## 1. Introduction

Diffuse glial tumors are the most common tumors of the central nervous system. Their prognosis depends on several factors. In 2016, the WHO introduced the notion of molecular biology into its classification of diffuse glial tumors in order to establish an integrated diagnosis [1]. Thereafter, this classification was based both on classical histological criteria and on molecular biology criteria, the most relevant of these being isocitrate dehydrogenase (IDH) 1 and 2 (mutated or nonmutated status), 1p/19q codeletion (codeleted or noncodeleted status), ATRX (loss or no loss status), and TERT (mutated or nonmutated status). The publication of the WHO 2021 classification saw the addition of CDKN2A or CDKN2B homozygous deletion as a new prognostic factor for IDH-mutated noncodeleted gliomas [2,3]. In the WHO 2016 and WHO 2021 classifications, Grade 2 and 3 oligodendrogliomas (ODs) are defined by the presence of mutations on the IDH 1 or 2 genes associated with codeletion of the chromosome arms 1p and 19q (1p/19q codeletion). These Grade 2 and 3 IDH mutant and 1p/19q codeleted ODs (OD2s and OD3s) are the diffuse gliomas associated with the best prognosis [4]. Favorable survival rates are linked to the nature of both the tumor and the response to different treatments, including radiotherapy and chemotherapy, especially the combination of procarbazine, lomustine, vincristine (PCV), and temozolomide (TMZ) [5,6,7]. When possible, the standard practice is to perform an extensive resection of low-grade gliomas based on several uncontrolled series showing that patients with no residual disease have the best survival outcome [8]. Although watch-and-wait strategies can be recommended for patients with positive prognostic factors such as complete resection, younger age, and OD2 [9,10], several randomized trials using PCV in addition to radiotherapy have established the benefit of this chemotherapy regimen in both OD3s [11,12] and OD2s [13]. The question of the best type of chemotherapy to use for OD2s or OD3s is still a matter of debate: TMZ, in addition to radiotherapy for WHO Grade 3 noncodeleted gliomas, showed a significant benefit in terms of survival for patients with IDH-mutant gliomas in the CATNON trial [14]. The ongoing ALLIANCE-N0577-CODEL prospective trial is currently assessing radiotherapy plus PCV versus radiotherapy plus TMZ [15]. 

However, despite the well-known sensitivity of ODs to radiotherapy and chemotherapy, the survival of patients with OD2s and OD3s is still heterogeneous [16]. CDKN2A homozygous deletion was recently shown to be associated with a poor prognosis close to that of glioblastomas for IDH-mutant gliomas lacking 1p/19q codeletion, as well as for OD3s [3]. However, although prognostic and predictive gene signatures have been established for many cancers, there are currently no such gene signatures for ODs. LMAN 1 has been associated with a better prognosis for 1p/19q codeleted gliomas compared with astrocytomas [17].

HOXA has been described as a prognostic factor for low-grade gliomas but in a mixed population that included astrocytomas and ODs [18]. In another example, increased expression of ESPL1 was shown to be associated with significantly shorter overall survival both in astrocytomas and in ODs, but with no analysis of treatment response [19].

Accordingly, by analyzing the clinical and molecular data of patients with IDH mutant and 1p/19q codeleted ODs who had been included since 2008 in the multicenter French national POLA cohort, our aim was to identify a gene signature of response to radiotherapy and chemotherapy for patients with OD2s or OD3s. 

## 2. Material and Methods

### 2.1. Patient Samples

Samples were obtained from patients included prospectively in the POLA network. All patients have given their written consent for clinical data collection and genetic analysis according to national policies. The study was approved by the ethics committee of the Hôpital Universitaire La Pitié-Salpêtrière on 3 October 2008. It was performed in accordance with the Declaration of Helsinki.

All patients were aged at least 18 years at diagnosis, and tumor histology was centrally reviewed and validated according to WHO guidelines [20]. We focused on Grade 2/3 1p/19q codeleted tumors treated at least by radiotherapy +/− chemotherapy.

### 2.2. RNA Extraction

Total RNA was extracted from frozen tumor samples using the iPrep™ ChargeSwitch™ forensic kit and the RNeasy Lipid Tissue Mini Kit (Qiagen). RNA integrity and quantity were assessed on the basis of the quality control criteria established by the Cartes d’Identité des Tumeurs research program (https://cit.ligue-cancer.net/). A 1-µg volume from each RNA sample was used to perform the gene expression analysis.

### 2.3. mRNA Expression Profiling and Analysis

The IGBMC Microarray and Sequencing Platform performed mRNA expression profiling using GeneChip^®^ Human Genome U133 Plus 2.0 arrays (Affymetrix, Santa Clara, CA, USA). We used the RMA algorithm (affy package) to normalize the data. Probe set intensities were then averaged per gene symbol. To reduce the number of predictors with low variance across the samples, only genes with an interquartile range above 1 were eligible for future variable selection.

### 2.4. Statistical Analyses

Demographic and clinicopathological data were subjected to the usual statistical analyses. Data were summarized by median and range for continuous variables and by frequency and percentage for categorical variables. Survival data were summarized using the Kaplan–Meier method. The Cox proportional hazards model was used for univariable analyses. Hazard ratios (HRs) were estimated with 95% confidence intervals (95% CI). We investigated enriched pathways using KEGG and Reactome databases to study the main pathways involved in OD response to radiotherapy and chemotherapy. 

A stability selection approach based on subsampling in combination with the lasso-pcvl algorithm was performed on genes significantly associated with progression-free survival in univariate analyses [21]. More specifically, the lasso-pcvl algorithm was applied to subsamples obtained by bootstrapping [22]. The proportion of subsamples in which a biomarker was selected corresponded to the selection probability for that biomarker. Only genes selected with a frequency equal to or above 0.6 were included in the final model. Finally, we calculated a risk score for prediction based on the linear predictor given by the multivariable Cox model. Risk groups (low-risk vs. high-risk) were obtained using time-dependent receiver operating characteristic (ROC) curves. A multivariable analysis was conducted to adjust the risk score according to clinical prognostic factors (age, presence of necrosis, and extent of resection) using the Cox model.

External validation was conducted on Grade 2/3 gliomas with 1p/19q codeletion treated with radiotherapy +/− chemotherapy from The Cancer Genome Atlas (TCGA) cohort.

All reported *p*-values were two-sided. For all statistical tests, differences were considered significant at the 5% level. Statistical analysis was performed using R. 3.4.2 software.

## 3. Results

### 3.1. Patient Characteristics and Survival Data

A total of 68 patients with 1p/19q codeleted ODs (67 OD3s, one OD2) treated with radiotherapy +/− chemotherapy were included in this study. Sample characteristics are set out in Table 1. Median follow-up time was 81 months (95% CI [75, 88]), and 12-, 24-, 36- and 48-month progression-free survival rates were 88.2 (95% CI [77.9, 93.9]), 70.6 (95% CI [58.2, 79.9]), 63.2 (95% CI [50.6, 73.4]), and 60.1% (95% CI [47.6, 70.7]). Median progression-free survival was 60.3 months (95% CI [41.0, not reached]). Twelve-, 24-, 36- and 48 -months overall survival rates were 98.5 (95% CI: [90.0–99.8]), 89.7 (95% CI: [79.6–95.0]), 88.2 (95% CI: [77.8–93.9]), and 86.7% (95% CI: [76.0–92.9]), respectively. Due to the low number of events, no further analysis was performed on overall survival.

### 3.2. Genes Associated with Progression-Free Survival

Filtering yielded 1697 genes with an interquartile range > 1. According to univariable analyses, 134 genes were associated with progression-free survival (*p* < 0.05). Among these 134 genes, 35 had overexpression associated with a longer progression-free survival (i.e., HR < 1, better prognosis) and 99 with a shorter progression-free survival (i.e., HR > 1, poor prognosis). When we explored significant Reactome pathways (Table 2), we found that genes involved in extracellular matrix organization, metabolism, and immune system pathways were the main ones associated with progression-free survival. 

### 3.3. Gene Signature of Response to Treatment

We applied the lasso-pcvl method combined with a resampling procedure to our cohort on the 134 genes associated with progression-free survival in the univariable analysis. The most stable predictors are listed in Table 3. Eight genes (ST3GAL6, QPCT, NQO1, EPHX1, CST3, S100A8, CHI3L1, and OSBPL3) whose overexpression was associated with shorter progression-free survival in univariable analysis (i.e., poor prognosis, HR > 1; Table 3) were selected in more than 60% of cases. These selected genes were then included in a multivariable model to calculate a risk score. This score was significantly associated with progression-free survival (HR = 2.72, 95% CI [1.85, 4.00], *p* < 0.001) and showed good calibration over time. The c-index was 0.75, and the area under the curve was 0.79, 0.77, and 0.78 at 12, 24, and 36 months, respectively. In the multivariable analysis, the eight-gene risk score remained statistically significant after adjustment on clinical prognostic factors (HR = 2.65, 95% CI [1.78, 3.95], *p* < 0.001). Risk groups were established (low-risk vs. high-risk) according to the ROC curve at 24 months (Figure 1). The c-index for risk groups was 0.61 (HR = 5.15, 95% CI [1.58, 16.84], *p* = 0.007). 

External validation was performed on the TCGA LGG cohort. We selected patients with 1p/19q codeletion treated by radiotherapy +/− chemotherapy (*n* = 69). The risk score was determined using the regression coefficients estimated from the POLA cohort, and its discriminative ability was confirmed on the validation set, with a c-index of 0.64. Among the nine patients who developed progressive disease within 24 months, seven were considered high-risk on the basis of the gene signature (time-dependent sensitivity at 24 months = 69.9%). 

## 4. Discussion

Our study was designed to highlight gene signatures associated with progression-free survival and response to radiotherapy and chemotherapy among patients treated for 1p/19q codeleted ODs. Two of the 68 patients we analyzed had 1p/19q codeleted tumors that were IDH wild-type and would not have been identified under the WHO 2021 classification as ODs. In all likelihood, these two patients actually had diffuse leptomeningeal glioneuronal tumors. Nevertheless, they were included in our study because they belonged to the POLA database established in 2008, and we decided to analyze the data of all 68 patients, as they all had 1p/19q codeletions. 

We found a gene signature composed of eight genes whose overexpression was associated with a shorter time to progression and which were mainly involved in metabolism, the immune system, and extracellular matrix organization. Epoxide hydroxylases (EPHX1) are Phase I xenobiotic detoxification enzymes that metabolize procarcinogens and are responsible for eliminating exogenous and endogenous compounds through oxidation reactions. Mutations in genes that lower their activity can lead to cellular DNA damage. Quinone oxidoreductase 1 (NQO1), a cytosolic reductase, is actively involved in the cellular response to many stresses. The cells are then protected by its upregulation against oxidative stress. It catalyzes the detoxification and reduction of quinine substrates [23]. NQO1 is overexpressed in many tumors, including glioblastoma, inhibiting oxidative stress and preventing cancer cell death. Its inhibition leads to in vivo EGFR-vIII positive glioblastoma growth inhibition [24]. NQO1 could therefore be a factor for aggressiveness in ODs. ST3GAL6 is a member of the sialyltransferase subfamily called ST3Gal. Its overexpression has been reported in multiple cancers, including breast cancer and hepatocellular carcinoma, and has been shown to be involved in urinary bladder cancer invasion and migration [25]. Its deregulation promotes cell proliferation and invasion through PI3K/AKT signaling, and it has been shown to be induced by HIF1α and IL6 or IL8 under hypoxic or inflammatory conditions [26,27].

Glutaminyl-peptide cyclotransferase (QPCT), a macrophage-specific gene, was recently included in a model predicting worse outcomes in patients with gliomas [28]. This model is a reflection of the tumor immune microenvironment and of tumor-associated macrophage (TAM) infiltration. Besides QPCT, chitinase-3-like 1 protein (CH3L1), which we also found to belong to our gene signature, has been demonstrated to modulate an immunosuppressive microenvironment in glioblastoma by reprogramming TAMs to M2-like phenotypes [29]. Innate and acquired immune responses also involve the S100A protein family, which regulates cell proliferation, migration, differentiation, and inflammation [30]. Several S100A members have been involved in glioma aggressiveness, such as S100A4, which regulates the epithelial-mesenchymal transition in glioblastoma and whose expression increases with grade [31]. S100A8 and S1009 proteins are potential immune modulators. In gliomas, a high expression of S100A8 and S100A9 inhibits T cell function and differentiation through interferon alpha to regulate macrophage or dendritic cell production [32]. In the literature, S100A8 expression has been shown to be associated with a worse prognosis in low-grade gliomas. Its expression appears to be more important during glioma grade progression [30]. Moreover, S100A9 has recently been shown to control brain metastasis radioresistance [33]. Thus, expression of S100A8 in ODs could reduce the efficacy of radiotherapy and chemotherapy by modulating these mechanisms. 

It appears that lipid metabolism, particularly reprogramming, has an important role in cancer cell growth, proliferation, angiogenesis, and invasion. OSBP is a family of oxysterol-binding proteins, of which oxysterol-binding protein-like 3 (OSBPL3) is a member. OSBPL3 is involved in lipid transport, lipid metabolism, and cell signaling by binding to phosphoinositides (PIP2 and PIP3) and interacting with the small GTPase R-RAS. In gastric cancer, the role of OSBPL3 has already been described: it promotes tumor growth by enhancing R-Ras/Akt signaling. It has been described as an independent biomarker of poor prognosis in this neoplasia. Overexpression of OSBPL3 has also been found and associated with poor prognoses in other tumors, such as colorectal, pancreatic, liver, bladder, and lung cancers, in the TCGA database [34]. In the BELOB trial, we note that OSBPL3 expression was related to a response benefit to dual therapy combining bevacizumab and CCNU in patients followed for recurrent glioblastoma. However, no clear mechanism was found to explain this response [35].

In our study, OSBPL3, which regulates lipid metabolism, is known to control glioma aggressiveness and appeared to be involved in proliferation through the Akt pathway in ODs. 

Our work highlighted an eight-gene signature associated with shorter progression-free survival and, thus, with therapeutic response in patients with IDH mutant and 1p/19q codeleted ODs, all of them treated at least with radiotherapy +/− chemotherapy. These genes, which modulate the immunosuppressive microenvironment and inflammation, metabolism, invasion, and extracellular matrix organization, seem to be major factors for poorer response to treatment and particularly to radiotherapy in ODs. Few studies have explored the heterogeneity of response to treatment for this tumor subtype, which is usually associated with better outcomes. Hu et al. identified a 35-gene signature of overall survival in patients with 1p/19q codeleted tumors, highlighting pathways involving N-terminal acetyltransferases, protein acetylation, response to copper ions, prostaglandins, and inflammation, all of which may be involved in 1p/19q glioma progression [36]. In their study, some patients underwent surgery but received no further treatment, while some also received radiotherapy, chemotherapy, or both. We only included patients who had been treated with radiotherapy at the very least and assessed progression-free survival in order to study heterogeneous sensitivity to radiotherapy in this 1p/19q codeleted tumor population. In both studies, Inflammation and metabolism seemed to be the main pathways involved in OD aggressiveness. 

Another study by the POLA Network involving the integrated analysis of the transcriptome, genome, and methylome revealed heterogeneity in 1p/19q codeleted tumors. It identified three subgroups of oligodendrogliomas with specific expression profiles of nervous system cell types: oligodendrocytes, oligodendrocyte precursor cells (OPCs), and neuronal lineage cells. More aggressive clinical and molecular features were found in the OPC subgroup, notably through MYC activation [37]. 

The cyclin-dependent kinase inhibitor 2A (CDKN2A) gene homozygous deletion has also been shown to be associated with dismal outcomes for IDH-mutant gliomas lacking 1p/19q codeletion, as well as for anaplastic ODs. Although we did not find CDKN2A homozygous deletion in our studied population, we did find a gene signature of shorter progression-free survival after radiotherapy +/− chemotherapy treatment, showing that independently of CDKN2A homozygous deletion, a subpopulation of patients with ODs have a more aggressive tumor that is less sensitive to treatment including radiotherapy. 

## 5. Conclusions

The present study performed on the national POLA Network database revealed a gene profile signature mainly involving microenvironment, immune, and metabolic pathways in patients with IDH mutant and 1p/19q codeleted ODs that was associated with a higher risk of rapid relapse after radiotherapy +/− chemotherapy. This new study confirms the heterogeneity of this population and should enhance the current understanding of ODs. This signature should also help clinicians identify patients who need more intensive treatment in order to conduct prospective trials of new treatments designed to inhibit these biological pathways. 

## Figures and Tables

**Figure 1 cancers-15-03067-f001:**
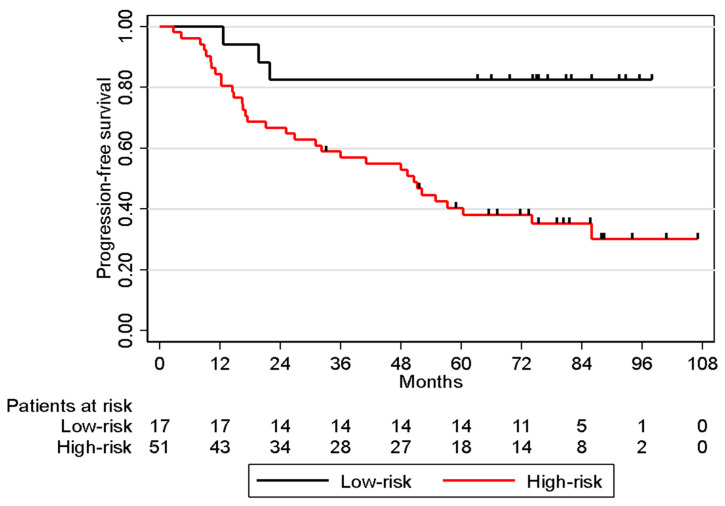
Kaplan–Meier curves for progression-free survival in POLA cohort stratified by eight-gene signature predictive of high or low risk.

**Table 1 cancers-15-03067-t001:** Patient characteristics.

		Patients *N* = 68	
		*n*	%
Age * in Years		45.7	23.7–64
Sex 68			
	Female	29	42.6
	Male	39	57.4
Tumor location 68			
	Bilateral	2	2.9
	Bilateral and median	2	2.9
	Left side	29	42.6
	Right side	35	51.5
Surgical resection 67			
	Complete	25	37.3
	Partial	17	25.4
	Subtotal	25	37.3
	Missing	1	
IDH mutation			
	IDH1R132H	58	85.3
	IDH2	8	11.8
	IDH wt	2	2.9
MGMT			
	M	26	68.4
	U	12	31.6
	Missing	30	
Presence of necrosis			
	No	50	73.5
	Yes	18	26.5
Treatment			
	PCV-RT	1	1.5
	RT	47	69.1
	RT-PCV	3	4.4
	Adjuvant RT-TMZ	2	2.9
	Concomitant RT-TMZ	3	4.4
	Stupp protocol	12	17.6

* median—range; IDH wt = IDH wild-type; M = methylated; U = unmethylated; PCV = procarbazine, lomustine and vincristine; RT = radiotherapy; TMZ = temozolomide

**Table 2 cancers-15-03067-t002:** Reactome pathways related to differentially expressed genes.

Pathway	Genes	*p*-Value	Corrected *p*-Value *
Neutrophil degranulation	BST2|CD58|CHI3L1|CST3|FTL|HLA-B|HLA-C|IQGAP1|NPC2|PECAM1|QPCT|S100A8|SERPINA1	9.94 × 10^−11^	5.37 × 10^−8^
Extracellular matrix organization	ACTN1|COL5A3|FBLN5|LAMB2|LTBP1|PECAM1|PLOD2|SERPINH1|TGFB2|VCAM1	1.74 × 10^−9^	9.42 × 10^−7^
Metabolism	ALDH1L1|ALDH6A1|DPYD|EPHX1|FAH|HMOX1|HSD11B1|IQGAP1|ITPKB|MAOA|MT1E|MT1F|MT1G|NPC2|NQO1|OSBPL3|PFKFB2|PIPOX|PPARGC1A|PSMB8|PTGS1|ST3GAL6	2.96 × 10^−9^	1.60 × 10^−6^
Innate immune system	ARPC1B|BST2|CD58|CHI3L1|CST3|FTL|HLA-B|HLA-C|IQGAP1|LYN|NPC2|PECAM1|PSMB8|QPCT|S100A8|SERPINA1|VWF	5.95 × 10^−9^	3.21 × 10^−6^
Immune system	ARPC1B|BST2|CD58|CHI3L1|CST3|FTL|HLA-B|HLA-C|HMOX1|IQGAP1|LYN|MAOA|NPC2|OSMR|PECAM1|PSMB8|QPCT|S100A8|SERPINA1|VCAM1|VWF	1.05 × 10^−8^	5.66 × 10^−6^
Platelet degranulation	ACTN1|PECAM1|SERPINA1|SERPING1|TGFB2|VWF	6.23 × 10^−7^	3.37 × 10^−4^
Response to elevated platelet cytosolic Ca^2+^	ACTN1|PECAM1|SERPINA1|SERPING1|TGFB2|VWF	7.75 × 10^−7^	4.19 × 10^−4^
Cytokine signaling in immune system	BST2|HLA-B|HLA-C|HMOX1|IQGAP1|LYN|MAOA|OSMR|PSMB8|VCAM1|VWF	1.36 × 10^−6^	7.36 × 10^−4^
Metallothioneins	MT1E|MT1F|MT1G	1.87 × 10^−6^	0.00101
Platelet activation, signaling, and aggregation	ACTN1|LYN|PECAM1|SERPINA1|SERPING1|TGFB2|VWF	4.03 × 10^−6^	0.00218
Hemostasis	ACTN1|CD58|LYN|PECAM1|PLAT|SERPINA1|SERPING1|TGFB2|VWF	1.03 × 10^−5^	0.00878
Interferon alpha/beta signaling	BST2|HLA-B|HLA-C|PSMB8	1.96 × 10^−5^	0.01056
Signaling by interleukins	HMOX1|IQGAP1|LYN|MAOA|OSMR|PSMB8|VCAM1|VWF	2.93 × 10^−5^	0.01584
Molecules associated with elastic fibers	FBLN5|LTBP1|TGFB2	4.92 × 10^−5^	0.02659
* Benjamini–Hochberg procedure			

**Table 3 cancers-15-03067-t003:** Occurrence frequency of genes selected in more than 60% of cases using the lasso-pcvl method.

Gene	Frequency	Unadjusted HR [95% CI]
ST3GAL6	0.68	1.62 [1.19, 2.20]
QPCT	0.68	1.62 [1.20, 2.19]
NQO1	0.68	1.74 [1.24, 2.45]
EPHX1	0.68	1.75 [1.23, 2.49]
CST3	0.67	1.69 [1.18, 2.41]
S100A8	0.65	1.56 [1.13, 2.17]
CHI3L1	0.64	1.40 [1.09, 1.81]
OSBPL3	0.61	1.56 [1.10, 2.21]

## Data Availability

Genes expression was obtained through the Cartes d’Identité des Tumeurs (CIT) national program from the Ligue Nationale Contre le Cancer accessed on (http://cit.ligue-cancer.net/) Grant « Prise en charge des oligodendrogliomes anaplasiques (POLA) Network», January 2012. Expression data are accessed on http://gliovis.bioinfo.cnio.es/; TCGA-LGG datasets are accessed on https://portal.gdc.cancer.gov/projects/TCGA-LGG.

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
