# Peer review of "A Multigene Signature Associated with Progression-Free Survival after Treatment for IDH Mutant and 1p/19q Codeleted Oligodendrogliomas"

_cancers, 2023, doi:10.3390/cancers15123067_

Round 1
Reviewer 1 Report
Overview: The authors conducted an analysis of the gene expression patterns in a cohort of patients with oligodendrogliomas with IDH mutations and 1p/19q codeletion. The authors were able to identify genes that were associated with PFS, better prognosis, and poor prognosis. They narrowed down the 99 genes observed to be associated with poor prognosis to 8 that were stable predictors, however the same was not done for the 134 genes associated with PFS or 35 genes associated with better prognosis. It also appears that the study focused only on genes whose overexpression was associated with poor prognosis, which could result in missing some crucial data on down-regulated genes. It is clear from the data presented in Figure 1 that these 8 genes are clearly associated with poor PFS. Please see below for a list of specific critiques labeled as either major or minor.
· The authors did not include data on any genes identified to be associated with progression-free survival (PFS) or better prognosis. The genes could provide critical information and allow for comparison of the gene signatures in the low-risk and high-risk groups. This should be included as Supplemental information. (Major)
· The authors did not perform the same analysis of the 134 genes associated with PFS or 35 with better prognosis that they did with the 99 genes associated with a poor prognosis to identify the most stable predictors. This analysis would significantly strengthen the overall study and contribute to being able to predict PFS. (Major)
· Table 2: Title is labeled as HR > 1 but in the text it is referenced as presenting the genes “associated with progression-free survival” (Line 67). Which is it? (Major)
· Table 2: representative genes from each pathway that were identified should be included as an additional column in the table. (Major)
· Lines 171-173: “genes… whose overexpression was associated with shorter progression-free survival…”. Was any analysis done to identify genes whose under expression or down regulation were associated with shorter PFS? The authors could be missing critical data in their gene signature if only focused on overexpression. (Major)
· Lines 278-278. The authors state they did not identify CDKN2A gene deletion in their study. However, if the authors are only analyzing for overexpression that this isn’t surprising since CDKN2A would be expected to show significantly lower expression. (Major)
· All of the tables would benefit from footnotes to define the acronyms used. For example: M and U for MGMT (presumably methylation status?), PCV-RT versus RT-PCV, in Table 1. In Table 2 explain how the corrected p value was calculated. (Minor)
· The HR term is never defined in the paper. (Minor)
· There is no reference included for the CATNON trial mentioned in line 80. (Minor)
· The authors do not define how they determined progression-free survival versus better prognosis groups. Some discussion of the cut-offs and how these were determined would be helpful in the methods. (Minor)
· While the authors state they did not include overall survival due to the low number of events, it would be interesting to see if there is any correlation between overall survival and the expression of the 8 genes identified. (Minor)
No issue with english language in paper, however please make sure to define acronyms used.
Author Response
We thank the reviewer for all the comments. Please, find our answers.
Comment 1 The authors did not include data on any genes identified to be associated with progression-free survival (PFS) or better prognosis. The genes could provide critical information and allow for comparison of the gene signatures in the low-risk and high-risk groups. This should be included as Supplemental information. (Major)
Response 1 : See response below.
Comment 2 The authors did not perform the same analysis of the 134 genes associated with PFS or 35 with better prognosis that they did with the 99 genes associated with a poor prognosis to identify the most stable predictors. This analysis would significantly strengthen the overall study and contribute to being able to predict PFS. (Major)
Response 2 : The analysis was performed on the 134 genes associated with PFS. Among these 134 genes, 35 were associated with a better prognosis (i.e. Hazard ratio (HR)<1) and 99 with a poor prognosis (i.e. HR>1). To avoid any ambiguity, this point has been clarified in the “Results” section.
Comment 3 Table 2: Title is labeled as HR > 1 but in the text it is referenced as presenting the genes “associated with progression-free survival” (Line 67). Which is it? (Major)
Response 3 : To avoid any ambiguity, the legend of the Table 2 has been updated.
Comment 4 Table 2: representative genes from each pathway that were identified should be included as an additional column in the table. (Major)
Response 4 : As asked, we added a column with the genes.
Comment 5 Lines 171-173: “genes… whose overexpression was associated with shorter progression-free survival…”. Was any analysis done to identify genes whose under expression or down regulation were associated with shorter PFS? The authors could be missing critical data in their gene signature if only focused on overexpression. (Major)
Response 5 : Please, see response above. The analysis was performed on the 134 genes associated with PFS (35 associated with a better prognosis (i.e. overexpression associated with longer PFS: HR<1) and 99 with a poor prognosis (i.e. overexpression associated with shorter PFS: HR>1)). Among these 134 genes, the eight genes selected by the lasso-pcvl method were associated with poor prognosis (i.e. overexpression associated with shorter PFS) in the univariable analysis.
This has been clarified in the “Results” section.
Comment 6 Lines 278-278. The authors state they did not identify CDKN2A gene deletion in their study. However, if the authors are only analyzing for overexpression that this isn’t surprising since CDKN2A would be expected to show significantly lower expression. (Major)
Response 6 : We studied the CDKN2A status but as written in the manuscript, no patient in our study presented a CDKN2A deletion.
Comment 7 All of the tables would benefit from footnotes to define the acronyms used. For example: M and U for MGMT (presumably methylation status?), PCV-RT versus RT-PCV, in Table 1. In Table 2 explain how the corrected p value was calculated. (Minor)
Response 7 : We added the Footnotes and explained in the table 2 how the corrected p value was calculated.
Comment 8 The HR term is never defined in the paper. (Minor)
Response 8 : The HR term has been defined in “Statistical analyses” section.
Comment 9 There is no reference included for the CATNON trial mentioned in line 80. (Minor)
Response 9 : As requested, we added the reference in the manuscript.
Comment 10 The authors do not define how they determined progression-free survival versus better prognosis groups. Some discussion of the cut-offs and how these were determined would be helpful in the methods. (Minor)
Response 10 : See response above. Determination of genes associated with better or poor prognosis was based on the value of the HR calculated in univariable analysis (i.e. HR<1 or >1).
Comment 11 While the authors state they did not include overall survival due to the low number of events, it would be interesting to see if there is any correlation between overall survival and the expression of the 8 genes identified. (Minor)
Response 11 : As stated in the manuscript, the low number of events did not allow us to perform further analyses on overall survival.
Reviewer 2 Report
The manuscript by Gilhodes et al. identified a signature of 8 genes as a marker of poor prognosis in IDH-mutant gliomas with 1p19q co-deletion.
As such data are rare, this study is worth of publishing to the field. However, the work is limited in the following aspects:
1. The work is entirely based on bioinformatic analysis. It appears that the anchorage in glioma pathogenesis is weak, though the authors have made efforts to discuss the potential relevance of the 8 signature genes.
2. Findings shown in Figure 1 should include the number of patients in each group.
3. How specific are the findings shown in Figure 1? In this sense, the effect of the 8-gene signature should be tested in IDH-mutant gliomas without 1p19q co-deletion and also in IDH-wildtype gliomas.
4. Would it be possible to replicate the findings in external datasets? How general is the applicability of the 8-gene signature?
Author Response
We thank the reviewer for all the comments. Please, find our answers.
Comments and Suggestions for Authors
The manuscript by Gilhodes et al. identified a signature of 8 genes as a marker of poor prognosis in IDH-mutant gliomas with 1p19q co-deletion.
As such data are rare, this study is worth of publishing to the field. However, the work is limited in the following aspects:
- The work is entirely based on bioinformatic analysis. It appears that the anchorage in glioma pathogenesis is weak, though the authors have made efforts to discuss the potential relevance of the 8 signature genes.
Response 1 : We thank the reviewer for this comment. We tried in the discussion to explain the biological role of each gene implicated in this signature and their potential involvement in OD aggressiveness.
- Findings shown in Figure 1 should include the number of patients in each group.
Response 2 : As asked, Figure 1 has been updated with the number of patients at risk in each group.
- How specific are the findings shown in Figure 1? In this sense, the effect of the 8-gene signature should be tested in IDH-mutant gliomas without 1p19q co-deletion and also in IDH-wildtype gliomas.
Response 3 : The question of this work that we asked was for oligodendroglioma , known to be of good prognosis but still heterogeneous in term of treatment response. We did not address the question on another disease as IDH wild type gliomas.
- Would it be possible to replicate the findings in external datasets? How general is the applicability of the 8-gene signature?
Response 4 : We validated the signature in an external data base as described in the results section.
Round 2
Reviewer 1 Report
The authors have improved the presentation of their data overall and clarified their methods. However, there are a couple of minor revisions still necessary to the manuscript.
Table 1: The authors indicated in their review response that they updated tables with footnotes to define the acronyms, however these footnotes are not showing up for Table 1 in the revised version provided by the authors.
Figure 1. The authors added patient numbers to figure 1, however I am confused how these relate to the graph. The low risk group only goes down to ~80% PFS but the number of patients goes from 17 to 0 after 108 months. The high-risk group also only goes to ~30% PFS but the number of patients goes from 51 to 0 after 108 months. Why are there 0 patients for both groups at 108 months? An explanation of this is needed.
Author Response
We thank the reviewer for all the comments. Please, find our answers.
Table 1: The authors indicated in their review response that they updated tables with footnotes to define the acronyms, however these footnotes are not showing up for Table 1 in the revised version provided by the authors.
Response Table 1 : This has been added under table 1 (see new manuscript version).
Figure 1. The authors added patient numbers to figure 1, however I am confused how these relate to the graph. The low risk group only goes down to ~80% PFS but the number of patients goes from 17 to 0 after 108 months. The high-risk group also only goes to ~30% PFS but the number of patients goes from 51 to 0 after 108 months. Why are there 0 patients for both groups at 108 months? An explanation of this is needed.
Response Figure 1 : Progression-free survival curves were built using the Kaplan-Meier method which takes censored data into account. We added on the graph the number of patients at risk in each group over time which corresponds to the number of patients who did not experienced the event of interest or were not censored before each time. For example, 1 and 2 patients were still at risk in Low and High-risk groups at 96 months. These 3 patients were then censored before 108 months (at 97.7, 100.5 and 106.8 months ; vertical marks on the curves), which means that there were no longer any patients at risk at that time.